# Genomic and Transcriptomic Characteristics of Esophageal Adenocarcinoma

**DOI:** 10.3390/cancers13174300

**Published:** 2021-08-26

**Authors:** Sascha Hoppe, Christoph Jonas, Marten Christian Wenzel, Oscar Velazquez Camacho, Christoph Arolt, Yue Zhao, Reinhard Büttner, Alexander Quaas, Patrick Sven Plum, Axel Maximilian Hillmer

**Affiliations:** 1Institute of Pathology, Faculty of Medicine and University Hospital Cologne, University of Cologne, 50937 Cologne, Germany; sascha.hoppe@uk-koeln.de (S.H.); christoph.jonas@uk-koeln.de (C.J.); mwenzel3@smail.uni-koeln.de (M.C.W.); oscar.velazquez-camacho@uk-koeln.de (O.V.C.); christoph.arolt@uk-koeln.de (C.A.); reinhard.buettner@uk-koeln.de (R.B.); alexander.quaas@uk-koeln.de (A.Q.); patrick.plum@uk-koeln.de (P.S.P.); 2Department of General, Visceral, Cancer and Transplantation Surgery, Faculty of Medicine and University Hospital Cologne, University of Cologne, 50937 Cologne, Germany; yue.zhao@uk-koeln.de; 3Center for Molecular Medicine Cologne, University of Cologne, 50931 Cologne, Germany

**Keywords:** esophageal adenocarcinoma, genomics, transcriptomics, mutation, tumor development, Barrett’s esophagus

## Abstract

**Simple Summary:**

Cancer of the esophagus is a deadly disease. There are two main subtypes, adenocarcinoma and squamous cell carcinoma, with adenocarcinoma of the esophagus (EAC) being more common in Western countries. Barrett’s esophagus (BE) describes a change in the esophageal surface near the stomach in response to reflux of gastric acid into the esophagus. BE increases the risk of developing EAC, and the incidence of EAC has risen dramatically over recent decades. One likely reason for the poor prognosis of EAC is based on the fact that each tumor has many genes affected by mutations, and most of these genes differ across patients, hampering the efficacy of therapies that target specific cancer driver proteins. In this review, we provide an overview of the gene mutations and gene activity changes in EAC and how these features can be used to divide patients into groups that might have different clinical characteristics.

**Abstract:**

Esophageal adenocarcinoma (EAC) is a deadly disease with limited options for targeted therapy. With the help of next-generation sequencing studies over the last decade, we gained an understanding of the genomic architecture of EAC. The tumor suppressor gene *TP53* is mutated in 70 to 80% of tumors followed by genomic alterations in *CDKN2A*, *KRAS*, *ERBB2*, *ARID1A*, *SMAD4* and a long tail of less frequently mutated genes. EAC is characterized by a high burden of point mutations and genomic rearrangements, resulting in amplifications and deletions of genomic regions. The genomic complexity is likely hampering the efficacy of targeted therapies. Barrett’s esophagus (BE), a metaplastic response of the esophagus to gastro-esophageal reflux disease, is the main risk factor for the development of EAC. Almost all EACs are derived from BE. The sequence from BE to EAC provides an opportunity to study the genomic evolution towards EAC. While the overlap of point mutations between BE and EAC within the same patient is, at times, surprisingly low, there is a correlation between the complexity of the genomic copy number profile and the development of EAC. Transcriptomic analyses separated EAC into a basal and a classical subtype, with the basal subtype showing a higher level of resistance to chemotherapy. In this review, we provide an overview of the current knowledge of the genomic and transcriptomic characteristics of EAC and their relevance for the development of the disease and patient care.

## 1. Introduction

Esophageal cancer is the eighth most common cancer worldwide, and the sixth most common cause of cancer-related death [1]. The most prevalent subtype of esophageal cancer in Western countries is esophageal adenocarcinoma (EAC). EAC has a poor prognosis, with an overall five-year survival rate of 20% and a median overall survival of less than a year due to the lack of targeted therapies and diagnosis usually made at late disease stages [1]. Supporting the importance of disease stage as a prognostic factor, EAC diagnosed at stage I has a five-year survival rate of 80%, decreasing to a dramatic 2% when diagnosed at stage IV, documented recently with a cohort from Northern Ireland [1]. Over the last decade, the incidence of EAC has risen dramatically in Western countries [2], emphasizing the need to improve therapeutic modalities for this tumor type with a poor prognosis. Neoadjuvant radiochemotherapy and perioperative chemotherapy are the standard of care, but the response varies dramatically, ranging from primary resistance to complete response. There are no robust biomarkers that allow predicting which patients will benefit from neoadjuvant therapy. Such biomarkers, however, would be very relevant for the clinic as they would spare non-responders from the toxic treatment. The genomic picture of EAC before and after neoadjuvant chemotherapy has been compared in a small number of studies [3,4,5]. Thus far, no common mechanisms for resistance could be identified.

The genomic architecture of EAC is rather complex, comprising many point mutations and genome structural alterations showing similarities to the chromosome instable group (CIN) of gastric cancer [6]. Overall, there is a need for the identification of patient subgroups based on a combination of molecular and clinical markers that allow a more tailored treatment of EAC patients. In this review, we provide an overview of our current understanding of the molecular characteristics of EAC based on systematic genomic and transcriptomic studies. We connect the molecular findings to clinical relevance and provide a perspective on what to expect in the near future.

## 2. Genomics of EAC

### 2.1. Somatic Variations and Copy Number Alterations

Whole genome sequencing (WGS) of a large cohort (*n* = 129) revealed that the genomic landscape of EAC is driven by large-scale genome alterations. Although point mutations are usually abundant, the recurrence of point mutations in specific genes across patients is low. Instead, Secrier and colleagues reported a dominance of genomic structural variations and copy number alterations [7]. The most frequently altered genes show a higher rate of rearrangements, amplifications or deletions, rather than insertions or deletions of a few base pairs (indels) or nonsynonymous point mutations. These rearranged genes comprise regulators of transcription, signaling and cell communication and include *SMYD3* (frequency of 39%), *RUNX1* (27%), *CTNNA3* (22%), *RBFOX1* (21%), *CDKN2A* and *CDKN2B* (18%) and *CDK14* (16%). Fragile sites such as *FHIT* (95%) and *WWOX* (84%) are also recurrently rearranged. Most tumors show evidence for mobile element insertions, with an average of 25 inserts per tumor. Frequently affected genes by mobile element insertions are *ERBB4* (5%), *CTNNA3* (4%), *CTNNA2* (3%), *CDH18* (2%) and *SOX5* (2%), all of which play important roles in regulating cell signaling, mitosis and adhesion. Amplified genome loci comprise the genes *ERBB2, EGFR, RB1, GATA4/6, CCND1* and *MDM2*. Deletions, which are generally less frequent, include *CLDN22* and *CDKN2A/2B*.

As it has been mentioned, point mutations and indels are abundant but heterogeneous, except for *TP53*. Nevertheless, seven genes reached significance in the analysis of Secrier and colleagues: *TP53* (81%), *ARID1A* (17%), *SMAD4* (16%), *CDKN2A* (15%), *KCNQ3* (12%), *CCDC102B* (9%) and *CYP7B1* (7%). Multiple receptor tyrosine kinases (RTKs) such as *ERBB2, EGFR, MET* and *FGFR* are often amplified in EAC tumors. Most samples (91%) have copy number gains in more than one RTK and/or amplifications in genes of the downstream MAPK and PI3K pathways. This may explain the poor results of the several current RTK therapy trials targeting single molecules of these pathways [8,9,10,11,12].

#### 2.1.1. Combining Methods and Expression Levels to Identify Driver Events in EAC

The high heterogeneity of EAC has driven the research community to combine cohorts and data types for systematic analyses with higher sensitivity to find common alterations among patients. Frankell and colleagues analyzed DNA sequencing data from a collection of 551 EACs in total, comprising WGS data of the International Cancer Genome Consortium (ICGC, *n* = 379) [13], WGS of Nones and colleagues (*n* = 22) [14] and whole exome sequencing (WES) of Dulak and colleagues (*n* = 149) [15], together with paired RNA sequencing (RNA-seq) data of 116 patients from the ICGC collection [13]. With a combination of methods detecting recurrent mutations, high-functional impact mutations, mutation clustering and amplification or deletion of genes that are over- or underexpressed, a total of 76 driver genes were suggested. The most frequently altered genes are *TP53* (72%), *CDKN2A* (28%), *KRAS* (19%), *MYC* (19%), *ERBB2* (18%), *GATA4* (15%), *SMAD4* (15%), *CCND1* (14%), *GATA6* (14%), *CDK6* (14%), *ARID1A* (13%) and *EGFR* (12%) (with *GATA4* and *GATA6* not meeting the general cancer genome criteria for oncogenes or tumor suppressor genes, respectively (https://cancer.sanger.ac.uk/census, accessed on 2 August 2021). The two driver events *GATA4* amplification and *SMAD4* mutation are prognostic for poor survival and therefore promising predictors for tailoring clinical care.

Non-coding driver elements discovered by ActiveDriverWGS, which considers functional impact, included regulatory regions of *TP53TG1, PTDSS1* and *WDR74*, which are described to be relevant for other cancer entities [16], and EAC-specific regions of the matrix metalloproteinase *MMP24, FTO, MTG2* and histone-encoding *HIST1H2BO* and *HIST1H2AM*.

Of 149 amplifications or deletions of genome loci detected by Genomic Identification of Significant Targets in Cancer (GISTIC), only 26% showed relative expression changes in the matched RNA-seq data, though the authors declared the low power to detect small expression differences. Among the altered genes were 17 known oncogenes including *ERBB2, KRAS* and *SMAD4*, emphasizing their relevance for EAC.

Ultrahigh copy number amplifications (ploidy-adjusted copy number > 10) are found frequently, likely due to extrachromosomal amplification via double minutes, and have a much stronger effect on expression changes than moderate copy number gains [13,14]. Naturally, but worth mentioning, tumor suppressor genes such as *ARID1B* are selected for truncating mutations, leading to dysfunctional isoforms, while oncogenes such as *ERBB2* are more prone to activating missense mutations. Interestingly, *TP53*-mutated tumors had more driver genes affected by somatic copy number changes than *TP53* wild-type tumors, in agreement with the protein’s prominent role as a key player for DNA integrity.

Underlining the importance of altered *TP53* function for EAC, *TP53* wild-type tumors are enriched for amplifications of *MDM2*, which degrades TP53, thus mimicking *TP53*-mutated tumors [4,13]. Dysfunctional *TP53* mutations and *MDM2* amplifications rarely appear in the same tumor, compatible with the equivalent functional consequences. Considering the participation of genes in cellular pathways, Frankell and colleagues found a mutation-driven activation of the Wnt pathway (19% of patients) and regulation of the SWI/SNF complex (28%) [13]. By a comparison of different pathways, the authors highlighted co-occurring mutations of *TP53* and *MYC, GATA6* and *SMAD4* and the Wnt and immune pathways, suggesting complementary roles for EAC pathophysiology. On the other hand, mutual exclusivity exists for a) *ARID1A* and *MYC*, b) pathways of RTKs and gastrointestinal differentiation and c) the DNA damage repair pathway and SWI/SNF, suggesting that the two members of each pair have similar roles in EAC pathophysiology. Using the Cancer Biomarkers database, the authors found >50% of EAC cases to be potentially sensitive to CDK4/CDK6 inhibitors, based on gene alterations in the related pathways, which could be validated by the first in vitro experiments [13].

#### 2.1.2. Large-Scale Sequencing of EAC Reveals EAC Helper Genes

The high heterogeneity of EAC with low frequencies of specific driver gene alterations makes it challenging to develop therapies. To overcome this problem, a machine learning algorithm called sysSVM has been used to detect recurrent but also rare altered genes contributing to cancer progression for individual patients [17]. Based on their similarity to known cancer genes regarding features including genomic alterations, protein network positions, miRNA associations, gene expression breadth and many more, a list of 952 helper genes has been established that might contribute to the progression of EAC. In accordance with the dominance of SCNAs in EAC, around 80% of the helper genes have copy number gains, accompanied by higher gene expression. Clustering by the proportion of perturbed pathways of the 952 helper genes divides the patient cohort into six subgroups with distinct molecular and clinical features. For example, EAC from subgroup 1 is enriched for the BRCA-related DNA damage response (DDR)-impaired mutational signature S3, and subgroup 4 is prognostic for poor survival. Experimental approaches show that alterations of helper genes increase EAC cell line growth to a similar extent as those of driver genes, underlining the clinical importance of helper genes. Of clinical interest is the fact that in vitro tumor cells develop a dependency on the altered helper genes, making them promising targets to develop patient-specific therapies. Interestingly, damaging alterations are also enriched in helper genes in pre-malignant Barrett’s esophagus (BE) samples and promote cellular proliferation in vitro. Thus, rare helper genes may contribute to the transition of BE to EAC, specifically for individual patients.

#### 2.1.3. Retrospective DNA Sequencing of Short Survivors

Hao and colleagues performed WES with treatment-naïve advanced (metastatic) EAC samples that were stratified into short (*n* = 20) and long (*n* = 20) survival [18]. The mutation burden is not different between short and long survivors. The most frequently mutated genes of the entire cohort were reported as *TP53* (70%), *SMAD4* (22%), *LRP1B* (20%), *FAT4* (18%), *ERBB4* (15%), *DMD* (15%), *CACNA1A* (12%), *SPTA1* (12%), *APC* (10%) and *PIK3CA* (10%). Deviating results from previous publications with larger cohorts are likely attributed to the fact that only tumors with the advanced stage 4b were included in this study. Notably, *KMT2C* alterations were found exclusively in short survivors and were associated with poor prognosis. Of note, short survivors had a higher intratumoral heterogeneity, with more subclones.

In addition to the known copy number losses on chromosomes 4, 9p, 17p and 18 and gains on chromosomes 20, 7p and 8q [6], loss of chromosome 4 was more frequent in the cohort of shorter survivors (95% vs. 40%) and was associated with poor survival in a larger TCGA cohort [18]. Tumors with chromosome 4 loss had a significantly decreased abundance of all major immune cell types, including T cells, suggesting a cold tumor type. Cold tumors are defined as non-immunogenic tumors without or with low infiltration of immune cells, especially T cells, making them invulnerable to immunotherapies.

### 2.2. Large-Scale Alterations and Genomic Catastrophes

The genomic landscape of EAC is characterized by large-scale alterations also known as genomic catastrophes. Genomic catastrophes can rapidly lead to alterations in many cancer-relevant genes driving cancer development. Chromotripsis is an event described by chromosomes shattered to fragments that are reassembled in a random order and orientation, resulting in a highly rearranged novel chromosome. The frequency of chromotripsis in EAC, at ~32%, is substantially higher compared to most other cancer entities and is accompanied by telomere shortening [7,14,19]. Besides substantial rearrangements of chromosomes, chromotripsis can generate extra-chromosomal double minutes harboring oncogenes. Amplifications in EAC resulting from such double minutes have been reported for oncogenes such as *MYC* and *MDM2* [14]. Chromotripsis in EAC can also lead to the loss of tumor suppressors such as *SMAD4* [14]. Interestingly, chromotripsis is associated with *TP53* inactivation [14,20]. Thus, the high incidence of chromotripsis in EAC is in accordance with the high recurrence of *TP53* mutations or TP53 inhibition by *MDM2* amplification. Murugaesu and colleagues found that chromotripsis, such as *TP53* mutation, is an early event in EAC [4].

Additional complex genomic events are present in 32% of EACs including kataegis (31%) and focal amplifications (<5 MB: 20%; 5–10 MB: 8%), the breakage-fusion-bridge (BFB) pattern (9%), double minute-like patterns (2%) and subtelomeric BFBs (1%) [7]. BFB results from mitotic failures during anaphase. Sister chromatids with shortened or lost telomeres fuse with one another and are pulled to the opposite poles of the cell by the mitotic spindle, eventually resulting in DNA breakage anywhere between the centromeres. The resulting chromatids again lack telomeres and are prone to repeated BFB cycles in the following cell divisions, which can lead to inverse duplications of genes. In EAC, BFB-driven amplifications were observed for oncogenes such as *MDM2, KRAS, RFC3* and *VEGFA* [14].

#### Loss of Y Chromosome

Loss of the Y chromosome (LoY) has been observed in various cancer types and even occurs in normal tissue of aging men. However, LoY is particularly frequent in EAC. Fluorescence in situ hybridization analysis of 400 male EACs including lymph node metastases revealed LoY in 52.5% [21]. Intriguingly, LoY was strongly associated with short overall survival, with 19.4 months for LoY and 58.8 months for male EAC patients with the Y chromosome. LoY was an independent prognostic marker but showed a correlation with *TP53* mutations, *KRAS* amplifications, loss of *ARID1A* and expression of *LAG3*. It remains unclear whether LoY contributes to the strong sex bias of EAC, with men being seven to nine times more frequently affected by EAC than females.

### 2.3. Mutational Signatures

In the area of cancer genomics, specific signatures of somatic genome alterations, particularly single-nucleotide variations (SNVs) and indels, have been identified that are described by the type of alteration, e.g., a C-to-T exchange, and the sequence context, i.e., the preceding and following bases [22,23]. Some of these signatures can be explained by specific carcinogens, e.g., tobacco smoke, demonstrating their value for the understanding of tumor development. Six mutational signatures are prominent in EAC tissues: S17A dominated by T > G substitutions in a CTT context known as the hallmark signature for EAC [15,24], a similar signature named S17B with additional T > C substitutions (note historic differences to the latest mutation signature nomenclature at https://cancer.sanger.ac.uk/signatures/, accessed on 2 August 2021), a complex pattern described as being caused by defects in the BRCA1/2 DNA repair pathway (S3), APOBEC-driven hypermutations of C > T in a TCA/TCT context (S2), age-related signature S1 described by C > T mutations in a 5′-CG dinucleotide context (* represents A, C, G, or T) and an S18-like signature with C > A/T substitutions in a GCA/TCT context [7,18]. Interestingly, DNA double-strand breaks described to be associated with mutational signature 3 are late-stage events of the disease [25].

Based on their dominant mutational signature, patients can be clustered into three subgroups called ‘C > A/T dominant’ (driven by S18-like and S1; 31% and 25% in the validation cohort, *n* = 87), ‘DNA damage repair (DDR)-impaired’ (S3; 15% and 22%) and ‘mutagenic’ (S17A and S17B, 53% and 53%) [7]. The DDR-impaired group has the highest genome instability, while the mutational and neoantigen burden is greatest in the mutagenic subgroup. An increase in defective mutations of genes in the homologous recombination pathway was detected in the DDR-impaired group, whereas the pathway’s key players *BRCA1* and *BRCA2* themselves are mutated in only 28% of patients within this subgroup. Nevertheless, Nones and colleagues demonstrated that EAC with extreme genomic instability can be driven by somatic *BRCA2* mutations accompanied by a prominent BRCA signature [14]. No clinical characteristics including tumor grade, response to therapy and survival are enriched in either of the three subgroups [7]. Instead, these subgroups might be used to stratify patients for specific therapies. Cultured cells with a dominant signature S3 (‘DDR-impaired’) are sensitive to treatment with the PARP inhibitor olaparib, but only when applied together with topoisomerase I inhibitor topotecan. Mitosis checkpoint inhibitors are most effective for cell lines with dominant mutagenic signatures S17A/B. These first results indicate the therapeutic relevance of mutational signatures in EAC, although further studies are needed for validation.

Two APOBEC-driven mutational signatures (S2, S13) are significantly enriched in the longer survival of patients [18]. Further analysis of The Cancer Genome Atlas (TCGA) data showed that higher degrees of the APOBEC signature in patients with elevated APOBEC3B mRNA expression indeed correlate with increased overall survival.

Tumors of patients who received chemotherapy or chemoradiation therapy have similar frequencies of the described mutational signatures compared to treatment-naïve samples. Comparing paired pre- and post-platinum-based chemotherapy tumor samples revealed that the prevalence of mutational signatures largely remains unchanged, except for a decreased proportion of C > T mutations and an increase in C > A mutations in a CpC context [4]. The latter signature is described as a response to platinum treatment, which is in accordance with the patient’s therapy in the described study. Nevertheless, the prominent signatures in EAC generally seem to persist after treatment and are described to be not associated with DNA-damaging chemotherapy or radiation therapy [25].

#### Instable Microsatellites Are Rare in EAC

Microsatellite instability (MSI) is a mutation signature caused by DNA mismatch repair defects associated with an extremely high load of indels. MSI tumors of a number of cancer entities have a better prognosis which is believed to be due to the larger number of neoantigens allowing the immune system to recognize and eliminate the respective cancer cells. The high level of neoantigens also explains the high response rates to immune checkpoint inhibitor therapies [26,27]. A total of 9 to 21% of gastric cancers and 4% of gastro-esophageal adenocarcinoma are MSI-positive [28,29]. For EAC in general, including more proximal cases (more distant to the gastro-esophageal junction), the frequency of mismatch repair-deficient tumors is only 0.65% [30]. While this is relatively infrequent, it is important to identify this group of patients in the clinic for considering immune checkpoint inhibitor therapy.

### 2.4. Comparing EAC with Esophageal Squamous Cell Carcinoma and Gastric Cancer

The TCGA research network published a comprehensive analysis of 164 esophageal carcinomas (72 EACs, 90 esophageal squamous cell carcinomas (ESCCs), 2 undifferentiated carcinomas) integrating WES plus shallow WGS, RNA-seq, miRNA sequencing, DNA methylation profiling and proteomics [6]. EAC could be clearly separated from ESCC based on expression profiles alone or together with other omics data layers, emphasizing the importance of considering these cancer types as distinct disease entities driven by cell lineage-specific alterations. Hypermethylation is much more frequent in EAC compared to ESCC. Recurrent mutated genes are rare but distinct for both groups, except for *TP53*. In addition, somatic copy number alterations are different with EAC-specific recurrent amplifications of *VEGFA, ERBB2, GATA6* and *CCNE1*, and deletions of *SMAD4*, all of which are absent in ESCC. RTKs and downstream mediators are frequently rearranged in EAC, led by *ERBB2* (32%), which is rearranged in only 3% of ESCCs. Dysregulation of the TGF-β pathway, activation of β-catenin and regulation of SWI/SNF complexes is common in EAC, but not in ESCC. Combination of the dataset with data from gastric cancers revealed that EAC is more similar to gastric carcinomas than to ESCC, while ESCC more closely resembles other squamous cell carcinomas [6]. All but one EAC were classified as chromosomal instable (CIN), resembling the CIN subgroup of gastric cancers. Separate analysis of EAC together with CIN gastric cancers showed that hypermethylation compared to normal tissue was strongest in EAC samples and significantly lower in gastric CIN, with decreased methylation towards distal gastric tumors. Hypermethylation in these cancers leads to epigenetic silencing of genes including *CDKN2A, MGMT* and *CHFR. MGMT* methylation increases the response to temozolomide treatment [31], and methylation of *CHFR* sensitizes tumors to dodetaxel and paclitaxel [32], as shown by experimental assays with esophageal cell lines. The authors concluded that EAC and CIN gastric cancers might be considered as a single disease named gastro-esophageal adenocarcinoma.

Next-generation sequencing of 592 target genes in a large cohort of tumors of the upper gastrointestinal tract (1176 EACs, 215 ESCCs, 1951 gastric adenocarcinomas) confirmed that EAC has a more similar molecular profile to gastric adenocarcinoma than to ESCC [33]. Some of the above-described recurrent mutations in EAC were significantly less frequent in ESCC, including *APC, ARID1A, CDH1, KRAS, PTEN* and *SMAD4*, while mutations of other genes such as *KMT2D, BAP1, CDKN2A* and *NOTCH1* were found less frequently in EAC. Only few differences were found between EAC and gastric adenocarcinoma, once more supporting the similarity of these adenocarcinomas and demonstrating that EAC and ESCC should be considered as distinct diseases.

### 2.5. Evolution before and towards Metastasis

As mentioned above, EAC is a highly heterogeneous cancer harboring multiple subclones. Spatial and temporal sampling followed by WES is a powerful tool to determine the genomic evolution of tumors. Somatic SNVs from these analyses are used to generate phylogenetic trees with trunks comprising early events, and branches resembling later stages of tumor progression. In EAC, the vast majority of oncogenic driver mutations appear in the trunks, including *TP53* and *CDKN2A* mutations [4]. Nevertheless, a remarkable amount of putative oncogenic mutations, such as *PIK3R1, SMAD4, TLR4* and *SLC39A12*, appear at branches, are subclonal and thus are later events. It is likely that the recurrence of such mutations is underestimated in reports that are based on the analysis of a single sample per patient. WES of multiple regions within one tumor showed that the majority of non-silent mutations can be detected in only fractions of the regions, with 55% of the mutations being intratumoral heterogeneous [4]. This once again demonstrates the difficulty of molecular diagnostics of EAC based on single small tissue samples, i.e., biopsies [4].

On the other hand, *TP53* mutations are clearly early events in EAC. Sequencing samples from different regions of one tumor usually displays clonality of *TP53* mutations accompanied by copy neutral loss of heterozygosity (LOH), resulting in the loss of the wild-type allele [4]. *TP53* mutations are chronologically followed by increased chromosomal instability (including chromotripsis) and genome doubling, both still early tumor-driving events. Polyploidy is a characteristic feature of EAC cells and recently was discovered to be derived from mitotic slippage due to faulty chromosome attachment to the mitotic spindle [34].

Interestingly, the EAC hallmark mutational signature S17 (T > G in a CTT context) is present at the early stages of tumor progression and decreases during the later stages [4]. S17 arises in cells exposed to gastric acid, and thus it is tempting to speculate that the mutational signature is a consequence of initial gastric reflux, a major risk factor for EAC, but thereafter does not strongly contribute to the progressive intratumoral mutational heterogeneity.

Spatiotemporal sampling of EAC patients with lymph node and distant metastases followed by WGS revealed that multiple subclones of the primary tumor spread to multiple regions in the body [25]. Cells from one subclone can spread to different tissue types, i.e., to the liver and lymph nodes, and one tissue type such as the liver can be infiltrated by cells derived from multiple subclones. Seeding at metastatic sites seems to occur rapidly, demonstrated by relatively few new SNVs with respect to the most recent common ancestor. Most of the recurrent driver SNVs as well as the most frequent CNAs described above appear in the primary tumor at an early time point before spreading [4,25]. Similarly, chromosomal amplifications are early events persisting in the later stages [4]. Nevertheless, after dispersion, genes of the subclones still undergo mutations and rearrangements including driver events such as *VEGFA* amplification and *MAP2K* mutation in some patients. Meanwhile, additional genomic alterations in the primary tumor appear, which can be distinct to the events at metastatic sites. The partially disparate genomic composition of metastatic sites and primary tumors emphasizes the problems in finding curative targeted therapies at an advanced stage of the disease. Notably, metastases show a genomic landscape similar to primary EAC with a dominance of copy number variations and large-scale alterations over somatic point mutations. Retrotranspositions of L1 mobile elements are more frequent in metastases, indicating an increased genomic instability. The mode of evolution where the primary tumor progresses to a state comprising multiple different subclones that are able to spread and colonize different or the same metastatic sites seems to be characteristic for EAC as the predominant mechanism and was named ‘clonal diaspora’.

### 2.6. Genomic Responses to Therapy

Chemotherapies and radiation therapies affect, in particular, highly proliferating cells as these therapies confer stress to the replication process and introduce DNA damage, resulting in increased DNA alterations after erroneous repair. Since EAC has a high rate of relapse after neoadjuvant chemoradiation or perioperative chemotherapy, it is of interest to understand therapy-induced mutations.

WES of 30 paired EACs before and after neoadjuvant chemotherapy with oxaliplatin-5-fluorouracil suggested that the tumors of most good responders pass through a genetic bottleneck resulting in a temporal loss of clonal diversity [3]. Even some poor responders showed evidence for passing through a bottleneck where the tumor regrew before surgery. In 20% of the analyzed tumors, the composition of *TP53* mutations changed after treatment, and in some situations, new driver gene mutations post-therapy have been observed.

WGS of ten matched tumors pre- and post-treatment with neoadjuvant chemotherapy combining a platinum-based agent, epirubicin and 5-fluorouracil, as recommended in the United Kingdom at that time, and a second cohort of 62 treatment-naïve and 58 chemotherapy-treated EACs showed no significant differences in SNVs, CNAs, mutational signatures or large-scale genomic alterations after therapy [5]. This is surprising, since these systemic drugs directly affect the DNA integrity and DNA repair mechanisms. The high heterogeneity of EAC with the evolution of distinct subclones within one tumor challenges the comparability of samples before and after treatment. Regions of the genome with LOH before treatment showed clear heterozygosity in samples after treatment, implying that sampling by chance occurred in distinct subclones that might have emerged from a common ancestor clone instead of having evolved from each other.

Similarly, Janjigian and colleagues used a capture-based next-generation sequencing approach to detect somatic mutations, copy number variations and rearrangements of a selection of cancer genes in patients with metastatic stage IV EAC before systemic chemotherapy and correlated the result with response to treatment [35]. Again, samples before and after therapy showed little divergence. No association between defects in homologous recombination-directed repair and survival was observed, and no alterations of known driver genes for homologous recombination repair defects, such as *BRCA1/2*, correlated with treatment response.

Targeted treatment of HER2 (ERBB2)-positive EAC with trastuzumab in addition to chemotherapy is the standard of care nowadays and results in a prolonged survival of these patients [35]. Overexpression of *ERBB2* is routinely discovered by IHC/FISH but can also be robustly detected by next-generation sequencing as it results from amplification of *ERBB2* [35]. Alterations of genes involved in the RTK/RAS/PI3K pathway in addition to *ERBB2* amplification are predictive for resistance to trastuzumab treatment. This illustrates the opportunity in the genomic characterization of patients to stratify them for specific treatment regimens. Interestingly, sequencing of paired tumors pre- and post-trastuzumab treatment revealed that 16% of patients lose the *ERBB2* amplification during disease progression upon treatment, illustrating a selection for an *ERBB2*-independent subclone as a resistance mechanism of trastuzumab-treated EAC. Furthermore, post-trastuzumab tumors had a deletion of *ERBB2* exon 16, resulting in a hyperphosphorylated ERBB2 isoform that is described to be resistant against ERBB2-targeted therapies in cancer models [35,36].

### 2.7. Genomic Evolution from BE to EAC

In the majority of cases, the development of EAC is triggered by gastro-esophageal reflux disease (GERD), which leads to a replacement of the squamous epithelium by a columnar epithelium [37]. The resulting still benign metaplasia termed Barrett’s esophagus (BE) increases the risk for EAC by 10- to 50-fold [38,39]. However, only 3.5% of individuals progress from BE to EAC during their lifetime [40]. It is therefore desirable to identify biomarkers for the progression of BE to EAC. An accepted model for the development of EAC is the following sequence of events: BE > low-grade dysplasia > high-grade dysplasia > EAC. Despite a benign histology, BE already often contains mutations that have been reported in cancer, e.g., alterations of *APC, CDKN2A* (p16) and *TP53* [41,42]. To understand the determinants of this sequence, several studies have investigated the genomic evolution of BE to EAC.

WGS of BE/EAC pairs of 23 patients displayed surprisingly little overlap between somatic SNVs identified in BE and paired EAC tissue of the same patient, with less than 20% overlap in 57% of the cases [42]. A higher overlap was observed between dysplastic BE and EAC. The study revealed an increase in somatic SNVs and copy number alterations in EAC compared to BE. This suggests that the progressive genomic instability that defines the clonal relatedness between BE and derived EAC is lower compared to the relatedness between dysplasia and EAC. In a similar study, analysis of WES of 25 BE/EAC pairs led to a model of two genomic trajectories of how BE transforms [43]: in trajectory 1, a gradual accumulation of alterations frequently affecting *TP53, CDKN2A* and *SMAD4* leads to dysplasia and genomic instability, where, finally, oncogene amplification results in EAC. In trajectory 2, early *TP53* loss is followed by genome doubling which, in turn, leads to genomic instability, oncogene amplification and EAC. Whole genome doubling in cancer is a mechanism that protects cancer cells from the deleterious effects of somatic alterations and allows for extensive LOH. These studies highlight the progressive genomic instability as a key mechanism for the development of EAC.

Martinez and colleagues investigated the genomic copy number profile of individual crypts and entire biopsies of four BE patients who progressed to EAC and of four patients who did not progress [44]. The study indicated that most BE segments (the entire part of the esophagus that is metaplastic) are clonal, with similar numbers and inferred rates of alterations for individual crypts and entire biopsies. Genome doubling and high levels of somatic CNAs were detectable in most individuals who later developed EAC four years before progression, whereas somatic CNA levels remained low in most non-progressors. Multi-region analysis suggested that BE forms from the clonal expansion of a single founder, rather than from polyclonal (trans-) differentiation of multiple lineages. This raises hope that the genomic analysis of biopsies can provide a representative genomic picture of BE, a prerequisite to stratifying patients into high- and low-risk groups, once robust genomic biomarkers can be established.

In an attempt to find such biomarkers, shallow WGS of 777 biopsies sampled from 88 patients in BE surveillance over a period of up to 15 years showed that genomic signals can distinguish progressive from stable disease even 10 years before histopathological transformation [45]. The selection of copy number profiles as a relatively abstract biomarker for progression to EAC reflects the difficulties to establish alterations of single genes for risk stratification. At the same time, genome-wide biomarkers provide an opportunity that should be considered in the clinical context.

The Oesophageal Cancer Clinical and Molecular Stratification Consortium integrated several ‘omics’ layers to define differences and similarities between BE and EAC. The consortium performed DNA methylation analyses of 150 BE and 285 EAC tissues and combined these data with transcriptome and genomic data, resulting in four molecular subtypes [46]. Subtypes 1 and 4 consisted almost exclusively of EAC, subtype 2 of BE and subtype 3 of both groups but enriched for EAC. Subtype 1 was characterized by DNA methylation, a high mutation burden and mutations in the cell cycle and RTK signaling pathway genes. Subtype 2 showed gene expression patterns associated with metabolic processes. Subtype 3 displayed no methylation changes compared to normal tissue but also immune cell infiltration, and subtype 4 was characterized by DNA hypomethylation associated with genome rearrangements and amplification of *CCNE1*. EAC cases of subtype 2 and subtype 3 had the highest and lowest survival probabilities, respectively, indicating that the molecular signature is of prognostic value.

The largest next-generation sequencing-based genome studies are listed in Table 1, and the main molecular characteristics of EAC are summarized in Figure 1.

## 3. Transcriptomics of EAC

The stratification of EACs into molecular subtypes by gene expression profiling methods provides new opportunities for understanding the molecular characteristics of EACs and developing possible therapeutic strategies. By analyzing gene expression profiling data of three independent EAC cohorts, two expression patterns could be defined [47]. Genes that were overexpressed in subtype 1 (basal subtype) were enriched in biological processes including epithelial cell differentiation, keratinocyte differentiation and the KEGG pathway basal cell carcinoma. Subtype 2 (classical subtype) showed a more similar expression pattern to BE. Correlating the subtypes with therapy response suggested subtype 1 to be more chemotherapy-resistant.

Integrating genomic and transcriptomic data of advanced EAC for risk stratification in the clinical context of 20 short vs. 20 long survivors, Hao and colleagues discovered novel molecular features for prognosticating overall survival [18]. The genomic analysis revealed alterations of the epigenetic modifier *KMT2C* exclusively in the short survivors together with a higher level of intratumor heterogeneity, whereas the APOBEC mutation signature was enriched in longer survivors. By clustering RNA sequencing data of 33 specimens of these patients, the authors identified three clusters, with cluster 1 mainly composed of tumors from long survivors and cluster 3 with tumors from short survivors. Tumors of cluster 1 showed a significantly increased expression of multiple immune-related markers such as *MPO, FCN1, CD200* and *LEF1*. Cluster 3 showed high expression of tumor promoters *MAP3K13, MECOM* and *JAK2*, predicting worse survival. *MAP3K13* upregulation has been reported to correlate with a poor outcome in tumor progression [48,49].

Highly significant expression changes in 17 known cancer genes such as *ERBB2, KRAS* and *SMAD4* were observed by analyzing the RNA sequencing data of 116 EACs, showing a correlation with a high degree of chromosomal instability [13]. The genomic landscape of driver events comprises mutations and CNAs in oncogenes and tumor suppressor genes. Copy number loss was not necessarily associated with a reduced expression of the tumor suppressor genes *ARID1A* and *CDH11* but instead was associated with loss of heterozygosity. The expression levels of *CDKN2A* compared to normal tissue suggest that *CDKN2A* is generally activated in EAC and returns to normal levels when deleted. Some genes showed overexpression or downregulation without genomic aberrations, for example, overexpression of *MYC*. *GATA4, GATA6* and *MUC6*, being involved in the differentiated phenotype of gastrointestinal tissue, were downregulated and may be lost during dedifferentiation observed in cancer [13].

### 3.1. RNA Sequencing of the Tumor Microenvironment in EAC

Li and colleagues focused, in their study, on characterizing the stroma microenvironment in a mixed cohort of EAC and ESCC, as an activated stroma and the extracellular matrix play a significant role in tumor initiation, progression and metastasis [50]. In their study based on previously published genomic and transcriptomic data with a training (*n* = 182) and a validation cohort (*n* = 227), the authors identified genes that were correlated with stromal elements. Based on their estimation of stromal activation, the authors could divide their cohorts into two subgroups, with subgroup 2 consisting of patients with high stromal activity, associated with a high tumor stage and increased stromal cell infiltration. Subgroup 2 showed worse survival. The identification of the stromal marker genes *MMP11, COL6A2, COL1A2, CTHRC1, FAP* and *LUM* that are involved in cancer-associated fibroblast (CAF) function highlighted the crucial role of CAFs in the tumor stroma of subgroup 2.

Changes in the immune profile during BE-to-EAC progression have been identified by RNA-sequencing of 65 BE/dysplasia/EAC samples [51]. A subset of chemokines and cytokines, in particular, IL6 and CXCL8, increased during BE progression to EAC. While high-grade dysplasia showed increased immune cell populations, EAC was immune-poor, with a rise in the immune checkpoint protein PD-L1 and a loss of CD8+ T cells providing a supporting rationale for recent efforts to investigate immune checkpoint inhibitor therapies for EAC.

### 3.2. Single-Cell RNA Sequencing of EAC

Single-cell transcriptomics is a major advancement in the field of RNA sequencing technologies. Conventional approaches such as bulk RNA sequencing lack the resolution to deconvolute the contributions of subpopulations that may be identified as cell types or states [52,53,54]. Using this technology, studying the tumor microenvironment (TME) on a single cell scale has been a focus of current research for many cancer entities, such as melanoma [55], breast cancer [56], lung cancer [57] and colorectal carcinoma [58]. The TME has been implied to play various roles in cancer progression, such as promoting metastasis, facilitating immune evasion, fostering angiogenesis and conveying therapeutic resistance [59]. By characterization of the different cell types and states within the TME, it is hoped to find new therapeutic approaches to halt cancer progression [59]. Several components of the TME of EAC have been found to correlate with patient outcomes. For example, CAFs have been reported to promote invasion by secreting periostin that activates the PI3K/Akt pathway by interaction with tumor cell integrins [60]. On the other hand, the infiltration of lymphocytes was found to be associated with improved cancer-specific survival as seen in other solid cancers [61].

Thus far, single-cell sequencing studies of the TME of EAC are pending. However, Owen and colleagues could demonstrate the accessibility of esophageal tissue to single-cell RNA sequencing [62]. It was possible to discriminate cell type differences in BE compared to normal tissue samples of the esophagus, stomach and duodenum, providing evidence of the similarity of BE and the esophageal submucosal glands. Furthermore, differentiated expression profiles of different cell types abundant in BE could be identified. This is of particular interest because of the association of BE with the genesis of EAC, as discussed above [62].

There is still a way to go in using single-cell RNA sequencing in patient diagnostics, since costs remain high and interpretation of results is time-consuming. A way to efficiently enhance the resolution of bulk RNA sequencing is ‘FACS-seq’. As we demonstrated previously, using a fluorescence-activated cell sorting (FACS) step before performing bulk RNA sequencing, it is possible to single out fibroblasts, leukocytes and epithelial or tumor cells. Thus, features of EAC and its tumor microenvironment become accessible to analysis in patient diagnostics, eventually extending patient stratification for personalized therapy options [63]. Major transcriptomic findings are summarized in Table 2.

## 4. Clinical Relevance of Molecular Characteristics

As illustrated above, EAC is a genetically complex disease with a high mutational burden and extensive chromosomal instability [13,17,64]. Remarkably, in EAC, not only alterations of classical driver genes such as *TP53, CDKN2A, SMARCA4, ARID1A, SMAD4, ERBB2* or *PIK3CA* but also those of helper genes such as *ABI2* and *NCOR2*, which are mutated only at a low frequency in single patients, contribute to tumor development [17,65,66].

In the context of such a heterogeneous disease, the successful transfer of genetic characteristics into clinical care is difficult and the prognosis of EAC remains devastating, resulting in an overall 5-year survival of about 20% [1,67]. Despite the development of multimodal treatment regimens based on a combination of either neoadjuvant chemoradiation [68] or perioperative chemotherapy [69] plus radical surgical resection, about one third of all patients show only little therapeutic response, with the majority of tumor cells still viable after neoadjuvant treatment [70]. Biomarkers are needed to identify patients at the time of diagnosis who are unlikely to benefit from therapy in order to spare them side effects. Interestingly, genome-wide sequencing approaches could demonstrate that almost half of the patients with EAC harbor somatic mutations in frequently alternated cancer pathways that are biomarkers or putative targets for therapy [15]. Nevertheless, only a few personalized therapeutic options based on specific molecular characteristics have made their way into the clinical routine.

Nowadays, patients with *ERBB2*-positive metastasized gastric or esophagogastric adenocarcinoma qualify for a combination of chemotherapy and the monoclonal cytotoxic *ERBB2* antibody trastuzumab, improving the patients’ survival compared to chemotherapeutic treatment alone [71]. Recent studies could demonstrate an even better outcome in such cohorts by adding immunotherapeutic treatment via PD-1 inhibition (monoclonal antibody: pembrolizumab) to this regime of *ERBB2* blockade and chemotherapy [72,73]. However, most data were derived from mixed cohorts of both entities, adenocarcinoma of a gastric or an esophagogastric junction origin. Of note, in large EAC-restricted cohorts, *ERBB2* positivity was associated with a survival advantage [74,75].

Other targeted therapies are still the subject of studies with ambivalent results. As 30–60% of metastasized patients with esophageal cancer have overexpression of *VEGFA* [76], the monoclonal antibody bevacizumab was utilized in combination with chemotherapy for first-line therapy in patients with adenocarcinoma of the esophagogastric junction (an entity overlapping with EAC) within the international AVAGAST study. Although the survival benefit was not significant, it demonstrated different ethnic therapy responses. While Asian patients did not respond at all, Caucasian populations showed marginally improved survival under treatment [10]. The combined *VEGF2/ERBB2* inhibitor ramucirumab has now been approved for second-line treatment of EAC, either as monotherapy or combined with chemotherapy such as Abraxane based on studies such as REGARD, RAINBOW or RAINFALL [77,78,79]. Inhibition of *EGFR* via monoclonal antibodies such as cetuximab or panitumumab in patients with esophageal cancer did not lead to improved survival in different analyses [9,11,12]. This approach has therefore not been considered as effective EAC therapy thus far. Similarly, *MET* inhibition via inhibition of its downstream target *HGF* with the antibody rilotumumab has not yet been successful in clinical studies [8]. Table 3 summarizes the current targeted approaches for EAC within the clinical routine.

Besides these current clinical limitations, knowledge of molecular characteristics provides an opportunity for future personalized treatments. Different mutational signatures of the tumors might result in deviating therapeutic regimens as Secrier and colleagues proposed [7]. Patients with dominant C > A/T mutational patterns associated with aging processes might benefit from conventional chemotherapy (in combination with ERBB2 inhibition) since the tumor’s genome is more stable. Patients with prevalent defects within those genes responsible for homologous recombination-mediated DNA repair might receive radiation combined with PARP inhibition as these tumors are vulnerable to DNA-damaging treatments and cannot repair the therapy-induced DNA damage sufficiently. Furthermore, the authors proposed another subgroup of patients with a dominant T > G mutational signature resulting in a high mutational burden and the presence of a high load of neoantigens. In these cases, immunotherapies such as checkpoint inhibition targeting CTLA4 and PD-1\PD-L1 might be promising [7].

Another central aspect of clinical interest is the estimation of putative tumor growth/shrinking under therapy. Besides classical histopathological features, e.g., lymphatic metastasis, locally advanced growth or tumor cell death under neoadjuvant therapy, molecular specifications detected in endoscopic biopsies and surgical specimens might help to improve risk stratification. Recent studies of combined strategies including targeted sequencing, screening for promoter methylation and WES in EAC patients revealed different response profiles associated with deviating histopathological features [4,80]. Increased copy number variations, mutations and amplifications of *CSMD1* or *ETV4* as well as CpG island promoter methylation correlated with a favorable histopathological response [81], while alterations in *SMARC4* or *SMURF1* were associated with a worse tumor response [81]. Interestingly, early during the evolution of EAC, genomic instability occurs and is relatively stable and preserved throughout the course under therapeutic pressure such as chemotherapy. Therefore, the resulting gene amplification may represent suitable candidates for targeted treatments [4]. However, such analyses and classifications have not made their way into the daily routine.

The upcoming clinical relevance of molecular tumor characteristics and usage of innovative genomic technologies is not only limited to therapeutic decisions and evaluation of treatment response but might also play an important role in early detection as well as surveillance of EAC and dysplastic BE as precursor lesions. Thus far, diagnostics rely on clinical parameters with esophagogastroduodenoscopy as the gold standard for detection since no other markers are available. Utilizing novel sequencing techniques provides an opportunity for primary diagnosis as well as surveillance during follow-up. The cytosponge technique, also known as ‘pill-on-a-string’, is a non-invasive method for the detection of abnormal epithelial cells in the esophagus. In 2014, Weaver and colleagues demonstrated that WGS and amplicon-based resequencing of 112 EACs, as well as 109 samples of patients with either non-dysplastic BE or high-grade dysplasia of the esophageal mucosa derived from cytosponge sampling, were able to identify significant transition points in the sequential development of the malignancy [24]. Somatic mutations within *TP53* only occurred in dysplastic BE and EAC, while *SMAD4* mutations indicated the transformation from dysplastic mucosa to early EAC. Therefore, this non-endoscopic approach might offer novel screening options for patients with dysplastic alterations at risk of early tumorigenesis. Recent analyses using the cytosponge technique strengthen these data, emphasizing the fact that the sampling bias of conventional endoscopic biopsies due to intratumoral heterogeneity and different tumor clones becomes obsolete [42,82]. Nevertheless, targeted sequencing of samples taken during routine endoscopic biopsy surveillance in BE patients was also able to identify progression to EAC based on the presence of *TP53* mutations. Progressors from dysplastic BE to EAC showed increased *TP53* mutations compared to ‘stable’ BE patients [83].

Finally, the transfer of knowledge gained from molecular characteristics as well as genomic profiling of EAC into clinical care is still unsatisfactory. However, more and more molecular data have become available, bearing a high potential for improved patient stratification. Therefore, the clinical importance of molecular characteristics in EAC will certainly continue to grow in the near future.

## 5. Conclusions

EAC is still a cancer with a poor outcome due to the very limited options for targeted therapies inhibiting RTKs and biomarker-based rationales for treatment concepts such as inhibitors of immune checkpoints, PARP and vascularization. It is likely that the lack of inhibitors of oncogenic drivers other than ERBB2 is due to the complex genomic architecture of EAC, with multiple putative driver alterations per tumor. The problem is illustrated by a model for smoking-related lung adenocarcinoma, where the smoking-induced high mutation load results in a high number of usually weak driver mutations in contrast to non-smokers with a low mutation load, who only develop lung cancer in the unlikely situation when one of the few mutations hits a strong driver such as *EGFR* [84]. It is plausible that a tumor that is driven by one strong driver against which a targeted drug is available responds better compared to a tumor that is driven by many (weak) drivers where inhibition of one of them only marginally reduces the overall pro-proliferative signaling. Therefore, a focus of current translational research on EAC may be on the identification of biomarkers that help to predict treatment response to therapies other than inhibitors of RTKs and related pro-proliferative signaling. Profiles such as the degree of genome instability, mutation load, stroma features and the immune repertoire, but also the germline genetic risk profile [85], should be systematically assessed and correlated with histopathological and clinical characteristics, particularly treatment response, to stratify EAC patients for more tailored therapies. Single-cell sequencing approaches are promising concepts to understand EAC biology and to identify cell type-specific biomarkers. Integrating the various data layers using machine learning provides an opportunity to define clinically relevant patient groups that cannot be easily described by one biomarker. Overall, we have gained an understanding of the EAC genomics over recent years and require integrative molecular concepts to translate molecular and clinical information into patient benefits.

## Figures and Tables

**Figure 1 cancers-13-04300-f001:**
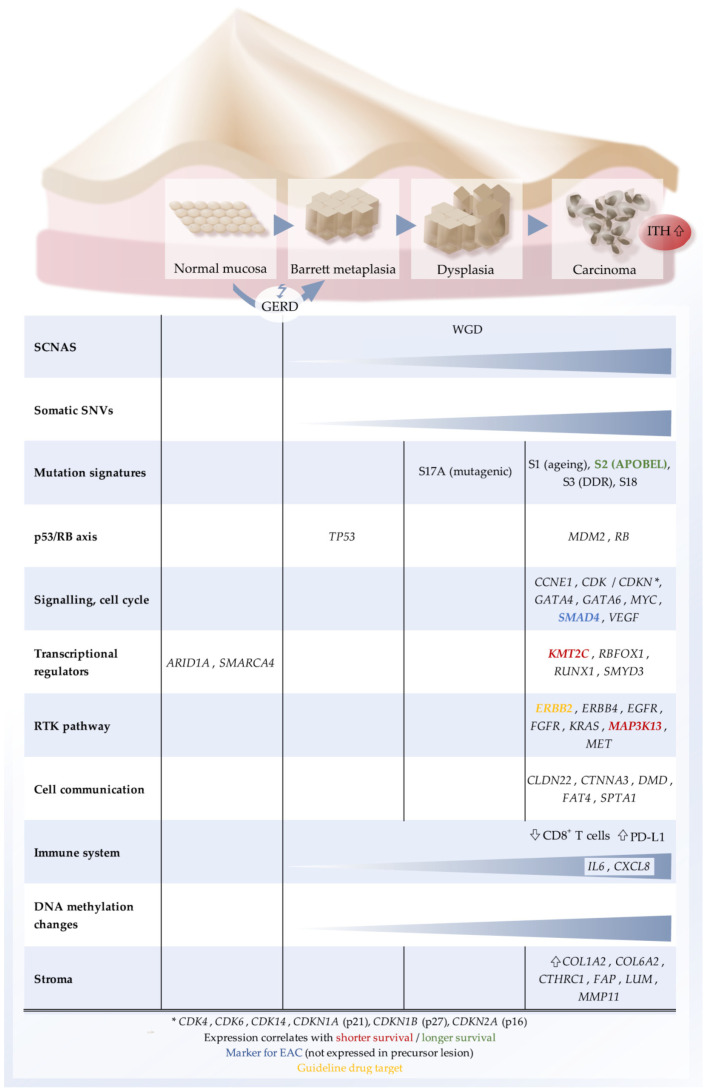
Schematic representation of characteristic molecular features of EAC. Top, sequence of histological states towards EAC. Bottom, list of molecular categories and key genes that are typically altered. GERD, gastro-esophageal reflux disease; ITH, intratumoral heterogeneity; RTK, receptor tyrosine kinase; SCNA, somatic genome copy number alteration; SNV: single-nucleotide variant; WGD, whole genome doubling.

**Table 1 cancers-13-04300-t001:** Main genome next-generation sequencing studies of EAC.

Sequencing Approach	Cohort	Main Findings	Study (Consortium)
WGS + SNP-arrays	22 EACs	- genomic catastrophies are frequent in EAC - chromotripsis appears in about one third of EACs, accompanied by telomere shortening - double minutes with oncogenes such as *MYC* and *MDM2* associate with chromotripsis in EAC - breakage-fusion bridge cycles are frequent and affect driver genes, i.e., *MDM2, KRAS* - extreme genomic instability can be driven by *BRCA2* mutations in EAC	Nones et al., 2014 [14]
WES	8 EACs, spatiotemporal samples *n* = 40	- many driver events are early, e.g., *TP53, CDKN2A*; some are late and subclonal, e.g., *PIK3R1, SMAD4* - genome doubling and chromosomal instability are early events leading to amplifications that persist through treatment - high heterogeneity associates with poor response to neoadjuvant chemotherapy	Murugaesu et al., 2015 [4]
WGS	129 EACs, including gastro-esophageal junction (Siewert types I and II)	- EAC landscape is highly heterogeneous, point mutations are abundant but with low frequency - dominance of SCNAs over SNVs - large-scale rearrangements are frequent, i.e., chromotripsis, breakage-fusion-bridge cycles - three molecular subtypes based on mutational signatures with therapeutic relevance	Secrier et al., 2016 (OCCAMS) [7]
WES + low-pass WGS + SNP arrays + DNA methylation profiling + mRNA-seq + miRNA-seq + proteomic	72 EACs (+ 90 ESCCs, 36 GEJs of unkown origin, 63 gastric GEJs, 140 gastric fundi or bodies, 143 gastric antral or pyloric, total 559)	- molecular features differentiate EAC from ESCC - EAC resembles chromosomal instable gastric adenocarcinoma - EAC is more commonly hypermethylated than ESCC or gastric cancers	TCGA, 2017 [6]
Targeted NGS	295 patients with metastatic esophageal, gastric and gastro-esophageal junction adenocarcinoma	- *HER2* positivity can be detected by NGS - alterations of genes involved in RTK/RAS/PI3K pathway in addition to *ERBB2* amplification are predictive for resistance to trastuzumab treatment - patients may lose *ERBB2* amplification or gain a resistant isoform during treatment	Janjigian et al., 2017 (MSK-IMPACT) [35]
WGS	matched pre- and post-therapy: 10 EACs, unmatched: 62 untreated + 58 treated EACs	- the genome of EAC treated with neoadjuvant chemotherapy is similar to untreated EAC - rare cases showing differences pre- and post-therapy arise from endoreduplication	Noorani et al., 2017 (OCCAMS) [5]
WGS + WES + RNA-seq	551 EACs: 379 ICGC (WGS) + 149 from Dulak et al. (WES) + 22 from Nones et al. (WGS). RNA-seq 116/379 ICGC	- combination of methods detects altered non-coding driver elements and driver genes, i.e., *TP53, CDKN2A, KRAS, MYC, ERBB2, GATA4, SMAD4, MMP24* - *TP53*-wt EACs have amplifications of *MDM2* - *SMAD4* and *GATA4* alterations predict poor prognosis - ~50% of EAC cases are potentially sensitive to CDK4/CDK6 inhibitors	Frankell et al., 2019 (OCCAMS) [13]
WGS + RNA-seq	267 EACs from ICGC	- identification of 952 helper genes contributing to cancer progression - helper genes identify six patient subgroups with distinct molecular and clinical features	Mourikis et al., 2019 (OCCAMS) [17]
WES + RNA-seq	40 EACs (stage 4b EGAC (Siewert types I, II and III)) divided into 20 shorter and 20 longer survivors	- similar mutational burden of short and long survivors - intratumoral heterogeneity is higher in short survivors - *KMT2C* alterations are exclusive for short survivors - loss of chr. 4 is associated with shorter survival	Hao et al., 2020 [18]
WGS	18 EACs, 388 spatiotemporal samples	- EAC progresses via clonal diaspora model with multiple different subclones of the primary tumor spreading and colonizing different or the same metastatic sites	Noorani et al., 2020 [25]

EAC, esophageal adenocarcinoma; EGAC, esophagogastric adenocarcinoma; ESCC, esophageal squamous cell carcinoma; ICGC, International Cancer Genome Consortium; MSK-IMPACT, Memorial Sloan Kettering-Integrated Mutation Profiling of Actionable Cancer Targets; NGS, next-generation sequencing; OCCAMS, esophageal cancer clinical and molecular stratification; SCNA, somatic copy number alteration; SNP, single-nucleotide polymorphism; SNV, single-nucleotide variant; WES, whole exome sequencing; WGS, whole genome sequencing.

**Table 2 cancers-13-04300-t002:** Major transcriptomic findings in next-generation sequencing studies of EAC.

**Sequencing Approach**	**Cohort**	**Main Findings**	**Study (Consortium)**
RNA-seq	Meta-analysis of three EAC cohorts: *n* = 88 (TCGA), *n* = 75 (GSE13898) and *n* = 52 (GSE19417)	Two molecular subtypes of EAC were identified based on RNA expression levels - Subtype I: basal cell-like; enriched for keratinocyte and epithelial cell differentiation; clusters with normal esophagus and gastric cancer; 24 specific genes mutated including *SMAD4*; less responsive to chemotherapy - Subtype II: classic EAC-like; clusters with dysplastic BE and ESCC; 30 specific genes mutated including *ARID1A*; more sensitive to chemotherapy	Guo et al., 2018 [47]
WES + RNA-seq	40 EACs (stage 4b EGAC (Siewert types I, II and III)) divided into 20 shorter and 20 longer survivors	Clustering of EAC samples based on RNA expression: - Long survivors associate with upregulation of immune-related markers, i.e., *MPO*, *LEF1* - Short survivors are enriched for upregulation of tumor promoters, i.e., *JAK2*, *MAP3K13* (please refer to Table 1 for genomics-related findings)	Hao et al., 2020 [18]
WGS + WES + RNA-seq	551 EACs: 379 ICGC (WGS) + 149 from Dulak et al. (WES) + 22 from Nones et al. (WGS). RNA-seq 116/379 ICGC	- 17 known cancer genes are frequently expressed high in EAC, i.e., *ERBB2, KRAS, SMAD4, MYC* - Copy number loss of tumor suppressors may not result in reduced expression, i.e., for *ARID1A* - *CDKN2A* is upregulated and returns to normal expression levels when deleted (please refer to Table 1 for genomics-related findings)	Frankell et al., 2019 (OCCAMS) [13]
RNA-seq	88 EACs + 94 ESCCs for training and 48 EACs (GSE19417) + 179 ESCCs (GSE53625) for validation	- Stratification of patients into two subgroups with high or low stromal activity - High stromal activity (subgroup 2, S2) was associated with high tumor stage and poor prognosis - S2 was enriched for EMT, angiogenesis and stromal infiltration of fibroblasts, endothelial cells and macrophages - Stromal marker genes characterizing S2 comprise genes important for CAF function	Li et al., 2020 [50]
RNA-Seq	65 patient samples: 25 non-dysplastic BEs, 29 high-grade dysplastic BEs, 11 EACs	- Chemokines and cytokines such as IL6 and CXCL8 increase during BE-to-EAC progression - Immune cell populations are high in dysplastic BE but low in EAC (i.e., CD8^+^ T-cells) - Immune inhibitory signaling, i.e., PD-L1 expression, is high in EAC, supporting trials with immune checkpoint inhibitors	Lagisetty et al., 2021 [51]
Single-cell RNA-seq	Total 4237 cells sequenced from 6 patients with BE + 2 patients with normal esophagus	- Distinct cell populations of BE are similar to submucosal gland cells (marker: *LEFTY1, OLFM4*) - Other cell populations are goblet cell-like and found in BE and colon samples (marker: *SPINK4, ITLN1*)	Owen et al., 2018 [62]
Flow sorting RNA-seq	Spatial sampling from 9 patients with EAC	- As an alternative to single-cell sequencing, specific cell populations from EAC can be labeled and separated using FACS followed by RNA-seq - Fibroblasts from EAC show upregulation of angiogenesis-related genes compared to fibroblasts from normal esophagus	Krämer et al., 2020 [63]

**Table 3 cancers-13-04300-t003:** Current approaches for targeted therapy in EAC (and other adenocarcinomas of the upper gastrointestinal tract) based on molecular characteristics.

Target	Treatment	Cohort Characteristics	Study
ERBB2	Trastuzumab	EAC + GEJ + Gastric	Bang et al., 2010 [71] (ToGA)
PD-L1	Pembrolizumab	ESCC + EAC + GEJ	Shah et al., 2019 (KEYNOTE-180 Study) [72]
PD-L1 + ERBB2	Pembrolizumab + Trastuzumab	EAC + GEJ + Gastric	Janjigian et al., 2020 [73]
VEGFA	Bevacizumab	GEJ + Gastric	Ohtsu et al., 2021 [10]
VEGF2 + ERBB2	Ramucirumab	GEJ + Gastric GEJ + Gastric GEJ + Gastric	Fuchs et al., 2014 (REGARD) [77] Wilke et al., 2014 (RAINBOW) [78] Fuchs et al., 2019 (RAINFALL) [79]
EGFR	Cetuximab	ESCC + EAC + GEJ	Huang et al., 2018 [9]
EGFR	Panitumumab	GEJ	Waddell et al., 2013 (REAL3) [12]
HGF	Rilotumumab	EAC + GEJ + Gastric	Catenacci et al., 2017 (RILOMET-1) [8]

EAC, esophageal adenocarcinoma; GEJ, adenocarcinoma of the gastro-esophageal junction; ESCC, esophageal squamous cell carcinoma; Gastric, gastric adenocarcinoma.

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
