# Peer review of "Genomic and Transcriptomic Characteristics of Esophageal Adenocarcinoma"

_cancers, 2021, doi:10.3390/cancers13174300_

Round 1

Reviewer 1 Report

This is the smartest and most comprehensive analysis of genomic analyses of esophageal adenocarcinoma and its precursor lesions, what these studies say about this cancer and therapeutic approaches, and what we are missing and still need to do.  The manuscript is beautifully written, captures the excitement and complexity of the problem in a manner that is unparalleled for this or any other cancer I have read.  The importance of this summary to the field cannot be overestimated, both for clinicians trying to understand how to apply the knowledge to a disease with clear heterogeneity in natural history, genomic profiles, and responses to therapies, and to scientists trying to glean unanswered questions in a field marked by iconic publications that seemingly answer everything but in fact are just a start.  Hoppe et al lays out a roadmap to key questions and never loses site of the fact that the diagnosis of EAC signals a bad outcome.    

Author Response

We thank the reviewer for the very positive assessment of the value of our review manuscript.

Reviewer 2 Report

Dear authors, I would like to thank you for the great amount of work that you have done. However there are some points that I believe would help the manuscript to improve.

  • I would suggest the authors to add more subheadings and define each subsection. The parts are sometimes too long and difficult to follow. In this way it would be easier for the reader to follow and find their subsection of interest.
  • I would also suggest to add more tables and/or figures. As mentioned above its quite difficult to follow hence by adding tables the key information could be summarized. many thanks  

Author Response

Point 1: I would suggest the authors to add more subheadings and define each subsection. The parts are sometimes too long and difficult to follow. In this way it would be easier for the reader to follow and find their subsection of interest.

Response 1: We thank the reviewer for this suggestion to improve the manuscript. We followed the recommendation and added a total of 7 subheaders to the manuscript. We also applied several changes at various paragraphs to define the sections. We agree that this will help the reader.

Point 2: I would also suggest to add more tables and/or figures. As mentioned above its quite difficult to follow hence by adding tables the key information could be summarized. many thanks 

Response 2: We agree that the topic is complex and might benefit from additional structured display items. We added two additional tables summarizing main findings based on expression analyses (Table 2) and relevant aspects for consideration in the clinical context (Table 3). We feel that this gives the reader a quick overview on main aspects on the topic.

Reviewer 3 Report

This manuscript comprehensively reviewed the previous reports and research regarding the genomic and transcriptomic characteristics of esophageal adenocarcinoma (EAC). As a reader, although lots of data was reviewed, this manuscript did not attract my attention and interest me. The authors just present others and colleagues’ report without personal comments on this manuscript which may be the big issue for current manuscript. In addition, the words used in the manuscript are not accurate so some sentences may confuse the readers.

Other comments:

  1. Introduction: EAC has a mortality rate of 85% and a median overall survival of less than a year due to the lack of targeted therapies and diagnosis, usually at late disease stages. It is not true as the prognosis of EAC depends on staging.

  1. 1. Somatic variations and copy number alterations. The frequency of point mutations and indels was low in the first paragraph but abundant in the second paragraph.
  2. ……a total of 76 driver genes were suggested. The most frequently altered genes are TP53 (72%), CDKN2A (28%), KRAS (19%), MYC (19%), ERBB2 (18%), GATA4 (15%), SMAD4 (15%), CCND1 (14%), GATA6 (14%), CDK6 (14%), ARID1A (13%) and EGFR (12%). Not all the genes listed here are driver genes.
  3. Intriguingly, LoY was strongly associated with short overall survival with 19.4 months for LoY and 58.8 months for male EAC patients with Y chromosome. What is the patient cohort? It remains unclear whether LoY contributes to the strong sex bias of EAC with men being 7 to 9 times more frequently affected than females. How did the females have LoY ?
  4. driver mutations such as TP53, CDKN2A, SMARCA4, ARID1A, SMAD4, ERBB2,

MYD88, PIK3CA, KAT6A or ARID2 contribute to tumor development. Not all of them are driver mutations.

  1. about one third of all patients do not show any therapeutic response at all. What is the definition of “no response at all” ?
  2. Without any benefits of the therapy, patients suffer from side effects. Patients always suffered from side effects regardless of side effects.
  3. About the discussion of trastuzumab in ERBB2-positive, some explanation is not true. The combination of trastuzumab and chemotherapy had better outcomes than those who received chemotherapy only limited in ERBB2-positive gastric cancer or ECJ cancer. In addition, the survival benefit is from treatment. Therefore, this is not associated with biology of ERBB2 as all the patients had the same genetic background. Second, pembrolizumab have shown better outcome in such cohort. Therefore, I suggest to revise this paragraph.
  4. The patients’ response towards multimodal treatment ….. ? what dose “the patients’ response” mean ?

Author Response

This manuscript comprehensively reviewed the previous reports and research regarding the genomic and transcriptomic characteristics of esophageal adenocarcinoma (EAC). As a reader, although lots of data was reviewed, this manuscript did not attract my attention and interest me. The authors just present others and colleagues’ report without personal comments on this manuscript which may be the big issue for current manuscript. In addition, the words used in the manuscript are not accurate so some sentences may confuse the readers.

Response

We carefully consider the reviewer’s comments. We indeed aimed at summarizing the main findings rather than writing a commentary. However, in the introduction and conclusion sections, we try to provide a higher level of interpretation. We hope that after addressing the points below, this review fulfils the reviewer’s expectations of an overview about omics-data from EAC samples.

Other comments:

  1. Introduction: EAC has a mortality rate of 85% and a median overall survival of less than a year due to the lack of targeted therapies and diagnosis, usually at late disease stages. It is not true as the prognosis of EAC depends on staging.

Response

Following the reviewer’s remark, we changed the paragraph as follows:

EAC has a poor prognosis with an overall five-year survival rate of 20% and a median overall survival of less than a year due to the lack of targeted therapies and diagnosis usually made at late disease stages [1]. Supporting the importance of disease stage as a prognostic factor, EAC diagnosed at stage I has a five-year-survival rate of 80% decreasing to dramatic 2% when diagnosed at stage IV, documented recently with a cohort from Northern Ireland [ 1].

  1. Somatic variations and copy number alterations. The frequency of point mutations and indels was low in the first paragraph but abundant in the second paragraph.

Response

We thank the reviewer for this comment. It is indeed the case, that point mutations in general are abundant within each tumor compared to other cancer entities. On the contrary, the recurrence/frequency of alterations in the same specific genes across patients are not abundant (except TP53). It is more the structural variations and copy number alterations that occur with high frequency for specific regions and genes. Unfortunately, we cannot find the mistake in our text. In the first paragraph of section 2 we write “Although point mutations usually are abundant, the recurrence of point mutations in specific genes across patients is low.”In the second paragraph we write:
“As mentioned, point mutations and indels are abundant but heterogeneous, except for TP53.”

  1. ……a total of 76 driver genes were suggested. The most frequently altered genes are TP53 (72%), CDKN2A (28%), KRAS (19%), MYC (19%), ERBB2 (18%), GATA4 (15%), SMAD4 (15%), CCND1 (14%), GATA6 (14%), CDK6 (14%), ARID1A (13%) and EGFR (12%). Not all the genes listed here are driver genes.

Response

We specified that these are not driver genes in general but driver genes for EAC, as stated by the original authors.

“[…] a total of 76 genes driving EAC were suggested.” We further clarified this by adding “(with GATA4 and GATA6 not meeting the general cancer genome criteria for oncogenes or tumor suppressor genes, respectively [https://cancer.sanger.ac.uk/census])

  1. Intriguingly, LoY was strongly associated with short overall survival with 19.4 months for LoY and 58.8 months for male EAC patients with Y chromosome. What is the patient cohort? It remains unclear whether LoY contributes to the strong sex bias of EAC with men being 7 to 9 times more frequently affected than females. How did the females have LoY ?

Response

The patient cohort was n=400 male EAC. We provided this information to the main text.
Our sentence raising the reviewer’s question how females have LoY was misleading. “men being 7 to 9 times more frequently affected” referred to the incidence of EAC in general, not the frequency of LoY. We rephrased the sentence:

“It remains unclear whether LoY contributes to the strong sex bias of EAC with men being 7 to 9 times more frequently affected by EAC than females.”

  1. driver mutations such as TP53, CDKN2A, SMARCA4, ARID1A, SMAD4, ERBB2, MYD88, PIK3CA, KAT6A or ARID2 contribute to tumor development. Not all of them are driver mutations.

Response

We shortened the sentence and reduced the list to actual driver genes. Also “mutations” was changed to “alterations of”, since also CNAs contribute to the list of these driver events (e.g. amplifications of CDKN2A).

  1. about one third of all patients do not show any therapeutic response at all. What is the definition of “no response at all” ?

Response

We changed the sentence to:

“[…] about one-third of all patients do show only little therapeutic response with the majority of tumor cells still viable after neoadjuvant treatment [70].”

  1. Without any benefits of the therapy, patients suffer from side effects. Patients always suffered from side effects regardless of side effects.

Response

We changed the sentence to

Biomarkers are needed to identify patients at the time of diagnosis who are unlikely to benefit from therapy in order to spare them side effects.

  1. About the discussion of trastuzumab in ERBB2-positive, some explanation is not true. The combination of trastuzumab and chemotherapy had better outcomes than those who received chemotherapy only limited in ERBB2-positive gastric cancer or ECJ cancer. In addition, the survival benefit is from treatment. Therefore, this is not associated with biology of ERBB2 as all the patients had the same genetic background. Second, pembrolizumab have shown better outcome in such cohort. Therefore, I suggest to revise this paragraph.

Response

We thank the reviewer for this comment. We revised the paragraph and changed it as follows:

“Nowadays, patients with ERBB2-positive metastasized gastric or esophagogastric adenocarcinoma qualify for a combination of chemotherapy and the monoclonal cytotoxic ERBB2-antibody trastuzumab [71] improving the patients’ survival compared to chemotherapeutic treatment alone. Recent studies could demonstrate an even better outcome in such cohorts by adding immunotherapeutic treatment via PD-1 inhibition (monoclonal antibody: pembrolizumab) to this regime of ERBB2-blockade and chemotherapy [72, 73]. However, most data were derived from mixed cohorts of both entities, adenocarcinoma of gastric or esophagogastric junction origin. Of note, in large EAC-restricted cohorts ERBB2-positivity was associated with a survival advantage [74, 75].”

  1. The patients’ response towards multimodal treatment ….. ? what dose “the patients’ response” mean ?

Response

We deleted “response towards multimodal treatment” and changed the beginning of this paragraph to:

“Another central aspect of clinical interest is the estimation of putative tumor growth/shrinking under therapy.”

Round 2

Reviewer 3 Report

the authors have fully response to my comments and I have no more questions.